# Antimicrobial use through consumption of medicated feeds in chicken flocks in the Mekong Delta of Vietnam: A three-year study before a ban on antimicrobial growth promoters

Nguyen Van Cuong[1], Bach Tuan Kiet[2], Vo Be Hien[2], Bao Dinh Truong[1,3], Doan Hoang Phu[1,3], Guy Thwaites[1,4], Marc Choisy[1,4,5], Juan Carrique-Mas[1,4]*

1 Oxford University Clinical Research Unit, Ho Chi Minh City, Vietnam, 2 Sub-Department of Animal Health and Production (SDAHP), Cao Lanh, Dong Thap, Vietnam, 3 Faculty of Animal Science and Veterinary Medicine, University of Agriculture and Forestry, HCMC, Ho Chi Minh City, Vietnam, 4 Nuffield Department of Medicine, Centre for Tropical Medicine and Global Health, Oxford University, Oxford, United Kingdom, 5 MIVEGEC, IRD, CNRS, University of Montpellier, Montpellier, France

* jcarrique-mas@oucru.org

**Data Availability Statement:** All relevant data are within the paper and its Supporting Information files.

## Abstract

Antimicrobials are included in commercial animal feed rations in many low- and middle-income countries (LMICs). We measured antimicrobial use (AMU) in commercial feed products consumed by 338 small-scale chicken flocks in the Mekong Delta of Vietnam, before a gradual nationwide ban on prophylactic use of antimicrobials (including in commercial feeds) to be introduced in the country over the coming five years. We inspected the labels of commercial feeds and calculated amounts of antimicrobial active ingredients (AAIs) given to flocks. We framed these results in the context of overall AMU in chicken production, and highlighted those products that did not comply with Government regulations. Thirty-five of 99 (35.3%) different antimicrobial-containing feed products included at least one AAI. Eight different AAIs (avilamycin, bacitracin, chlortetracycline, colistin, enramycin, flavomycin, oxytetracycline, virginamycin) belonging to five classes were identified. Brooding feeds contained antimicrobials the most (60.0%), followed by grower (40.9%) and finisher feeds (20.0%). Quantitatively, chlortetracycline was consumed most (42.2 mg/kg SEM ±0.34; 50.0% of total use), followed by enramycin (18.4 mg/kg SEM ±0.03, 21.8%), bacitracin (16.4 mg/kg SEM ±0.20, 19.4%) and colistin (6.40 mg/kg SEM ± 4.21;7.6%). Other antimicrobials consumed were virgianamycin, avilamycin, flavomycin and oxytetracycline (each ≤0.50 mg/kg). Antimicrobials in commercial feeds were more commonly given to flocks in the earlier part of the production cycle. A total of 10 (9.3%) products were not compliant with existing Vietnamese regulation (06/2016/TT-BNNPTNT) either because they included a non-authorised AAI (4), had AAIs over the permitted limits (4), or both (2). A number of commercial feed formulations examined included colistin (polymyxin E), a critically important antimicrobial of highest priority for human medicine. These results illustrate the challenges for effective implementation and enforcement of restrictions of antimicrobials in commercial

**Funding:** JCM was funded by Wellcome Trust (Grant Reference Number 110085/Z/15/Z).

**Competing interests:** The authors have declared that no competing interests exist.

feeds in LMICs. Results from this study should help encourage discussion about policies on medicated feeds in LMICs.

## Introduction

Antimicrobials are used in veterinary medicine. The global annual consumption of antimicrobials intended for animal use has been estimated in the region of 63 thousand tonnes [1]. In European Union (EU) countries, all of which have antimicrobial consumption surveillance systems based on sales data, antimicrobials intended for animal use quantitatively represent approximately 2/3 of total antimicrobial use (AMU) [2]. It is believed that excessive use of antimicrobials in animal production is one factor contributing to the global rise in antimicrobial resistance (AMR), although its magnitude is unknown [3, 4]. The total amounts of antimicrobials intended for animal production are expected to increase in coming years due to intensification of livestock production, mostly in low- and middle-income countries [1].

Antimicrobials are used in veterinary medicine to treat and prevent animal disease. In addition, in many countries they are also added to feed rations in sub-therapeutic concentrations in order to increase animal growth and productivity (antimicrobial growth promoters, AGPs). Their mechanism of action is however, poorly understood [5].

Over the last years, the issue of AMR and excessive AMU has attracted considerable attention worldwide, and many policy initatives aimed at reducing AMU/AMR have been recently developed by global organizations such as the World Health Organization (WHO) [6], the Food and Agriculture Organization of the United Nations (FAO) [7], and the World Organization for Animal Health (OIE) [8].

Antimicrobials in feed (including AGPs) have been the subject of much debate over recent years. Those opposing banning/restrictions of AGPs often express concerns based on potential losses in productivity, as well as the likelihood of emergence of certain diseases (i.e. necrotic enteritis in chickens) [9]. Positions in favour of their restriction often align themselves with the need to protect the efficacy of antimicrobials for human health. In the European Union (EU), mostly because of public health pressure, AGPs were banned in 2006 [10]. In recent years, and in line with FAO recommendations [7], some countries have started implementing bans or restrictions on AGPs in animal feeds, In the USA, phasing out of certain AGPs commenced in 2013 [11] and currently medically important AGPs are not allowed. In the Asia-Pacific region, countries such as Korea (2011), Australia (2013) [12] have implemented bans of AGPs in animal feeds. Other countries such as Thailand (2015) [13], China (2016) [14] and India (2019, Ministry of Health) also subsequently adopted policies that restrict AGPs in commercial feeds.

Worldwide annual consumption of poultry meat (2013–2015) stood at 110,280 tonnes, second only to pork (117,005 tonnes). By 2025, chicken production is expected to surpass that of pork production [15].

In Vietnam, antimicrobials are often found in both commercial pig and poultry rations. A study estimated in-feed consumption of antimicrobials extrapolated from a retail survey of commercial feeds in 77mg of AGPs per kg of chicken produced [16]. A study of medium-sized chicken farms estimated that chickens consumed 57mg of AGPs per kg of animal produced [17]. However, that study was based on a small sample of 6 farms.

A 2002 Vietnamese government regulation on animal feeds (54/2002/QĐ-BNN) included a ban on 18 chemicals (including chloramphenicol, metronidazole and nitrofurans). Further

(2014), legislation (28/2014/TT-BNNPTNT) expanded this list to bacitracin, carbadox and ola-quindox. In May 2016 Vietnam issued Circular 06/2016/TT-BNNPTNT, listing those antimicrobial active ingredients (AAIs) authorized for inclusion in commercial feed types as AGPs, as well as the maximum levels allowed in each feed type. According to this regulation, the maximum number of different AAIs to be included in each feed was limited to two. In 2018, Vietnam introduced its Animal Husbandry Law (32/2018/QH14) generically banning the use of AGPs in commercial feeds. A further Decree (13/2020/ND-CP) included the timeframe for a ban on AMU for prophylactic purposes (including AGPs), with phased bans for different antimicrobials classes: WHO 'highest' and 'high priority' critically important AAIs to be banned from 2021, highly important AAIs from 2022, important AAIs from 2023 and all other antimicrobial classes from of 2026 [18].

This study aimed at investigating the types and quantities of AAIs in commercial feed in a large representative cohort of small-scale chicken flocks in the Mekong Delta region of Vietnam immediately before the implementation of the new decree. This information complements existing data on antimicrobials administered in water by the farmer [19], and provides the full picture on antimicrobial consumption in small-scale commercial chicken flocks in the area. This knowledge should form the basis of informed decisions aiming at reducing AMU in chicken production in Vietnam.

## Materials and methods

### Farm selection

Farm owners in two districts (Cao Lanh, Thap Muoi) within Dong Thap (Mekong Delta, Vietnam) were randomly selected from the official (Sub-Department of Animal Health) farm census and were contacted by the veterinary authorities. Farmers about to start raising flocks of ≥100 chickens using native breeds that practiced all-in/all-out management were recruited, and flocks were followed up longitudinally. A total of 115 farms were recruited (59 in Cao Lanh; 66 in Thap Muoi). This study was performed in the context of a large field based trial aimed at reducing AMU in chicken production through the provision of veterinary advice [20]. Owners of selected farms were requested to record in detail the types of commercial feed used and to keep the sacs/containers of all feed products used. A field study team visited farms four times over the production cycle to collect data on commercial feed products used by week. A total of 338 flocks raised in these farms were investigated. Of the 115 farms, 44 completed 1 cycle (38.3.4%), 25 (21.7%) 2 cycles, 8 (7.0%) 3 cycles, 11 (9.6%) 4 cycles, 12 (10.4%) 5 cycles, and 15 (13.0%) more than 5 cycles. The median flock size at restocking was 303 [IQR 200–500]. A total of 6,041 weeks of data were collected. The median duration of these native meat chicken production cycles was 19 [IQR 17–21] weeks. All farm visits were conducted from October 2016 to Oct 2019.

### AAIs in commercial feed products

All commercial feed products containing an antimicrobial active ingredient (AAI) were singled out after inspecting their label. AAIs were described by: (1) target species (duck, chicken or pig); (2) indication by stage of production (brooder, grower or finisher); and (3) type of formulation (crumbs, mash or pellets). From each feed product, we described the AAIs contained and their concentration (expressed in mg/kg product). AAIs were classified based on the OIE list of antimicrobial agents [21] and any antimicrobials regarded as critically important by WHO [22] were highlighted. We excluded ionophore coccidiostats (aimed at controlling coccidial infections) since it is thought that these substances do not have a link with resistance against antimicrobials commonly used to treat human or animal bacterial disease. We

identified those feed products containing antimicrobials at concentrations not permitted under Vietnamese legislation [23].

## Data analyses

We calculated consumption of each AAI included in commercial feed by week by relating the amounts of AAI (mg) to the weight of birds at the time of consumption (standard weight of the flock) (kg) (mg/kg live chicken) for all weeks (*n*) over the flock's life duration (Expression 1).

$$\text{mg/kg chicken at time of consumption} = \sum\nolimits_{k=1}^{n} \frac{AAI\ used\ (mg)\ in\ week\ k}{Standard\ weight\ of\ the\ flock\ (kg)\ at\ week\ k}$$

Weekly consumption of AAIs in feed was calculated by multiplying weekly feed consumption by the AAIs concentration indicated in that feed. The feed consumption was estimated from unpublished data related to native Vietnamese layer pullets, where 443g of feed were consumed by 1 kg of live chicken per week. The denominator (total weight of the flock at week *k*) was calculated from the number of chickens present in the flock multiplied by an estimated (standard) weight. The latter was based on weekly weight data from 10 randomly selected chickens from 11 representative flocks, collected from week 1 until week 22 of their production cycle [19].

The concentration (strength) of AAI in each feed product was obtained from its label. However, information for a number of feed products contained uncertain information in their labels, concerning the identity of the AAI and the amounts included. For feed products with AAI content ambiguously labeled (i.e. indicating inclusion of one of >1 listed AAIs), the amount of each AAI was calculated by assigning each antimicrobial a probability corresponding being included (probability = 1), and not being included (probability = 0). For products indicating their AAIs concentration as a range, lowest and highest estimates were calculated for each antimicrobial. The amounts of each AAIs were summarized in each flock by AAI and by week.

## Ethical committee approval

This study was granted ethics approval by the Oxford Tropical Research Ethics Committee (OXTREC) (Ref. 5121/16) and by the local authorities (People's Committed of Dong Thap province). All participating farmers provided written consent.

## Results

### Description of commercial feed products

A total of 99 different commercial feed products were identified. Those products were intended for chicken (85 products, 85.9%), pig (12, 12.1%), and duck (2, 2.0%) feeding. Feed products intended for chickens were classified according to their indication (production stage): 38 for brooding, 22 for growing (i.e mid-production) and 25 for finishing. Of those, 35 (35.3%) (all intended for chicken use) contained at least one antimicrobial. A total of 21/38 (60.0%), 9/22 (40.9%) and 5/25 (20.0%) commercial feeds intendend for brooding, growing and finishing, respectively, contained antimicrobials. Detailed information on all antimicrobial-containing feed products is available in S1 Table. All except one product (a brooder feed that contained both chlortetracycline and colistin) contained one AAI. A total of 12 (34%) products had an ambiguous label, indicating that it contained one of 2–4 listed AAIs. A total of 8 different AAIs belonging to 5 classes were listed in the 35 feed products. The most

**Table 1. Antimicrobial Active Ingredients (AAIs) and their concentrations in 85 chicken feed products given to flocks in Dong Thap (Mekong Delta, Vietnam).**

| AAIs | Class | Products (n = 85) (%) | AAI mean concentration [range in mg/kg feed] (No. products) | | | **Permitted concentration [range in mg/kg feed] |
|---|---|---|---|---|---|---|
| | | | **Brooder** | **Grower** | **Finisher** | |
| Enramycin | Polypeptides | 16 (18.8) | [7.7–10] (7) | [8.2–10.0] (5) | [11.6–11.6] (4) | [1–10] |
| Bacitracin | Polypeptides | 14 (16.5) | [51.1–63.1] (8) | [125.0] (1) | [50–60] (5) | [4–50] |
| Chlortetracycline | Tetracyclines | 13 (15.3) | [52.7–61.1] (9) | [40.0–50.0] (4) | - | [10–50] |
| Avilamycin | Orthosomycin | 5 (5.9) | [12.5] (2) †† | [15.0] (1) | [10.0] (2) †† | NAA |
| Flavomycin | Other† | 5 (5.9) | [6.0] (2) †† | [2.0] (1) | [10.0] (2) †† | NAA |
| Colistin* | Polipeptides | 4 (4.7) | [70–136.6] (3) | - | [60.0–160.0] (1) | [2–20] |
| Virginamycin | Streptogramin A | 2 (2.4) | [5.0–15.0] (1) | - | [5.0] (1) †† | [5–15] |
| Oxytetracycline | Tetracyclines | 1 (1.2) | [50.0] (1) | - | - | NA |

*Critically important antimicrobial class according to WHO.

**AAIs permitted in chicken feeds from 1 to 28 day old birds (brooder and grower feeds) [24]. NAA = Not allowed antimicrobial.

†Antibiotic complex obtained from *Streptomyces bambergiensis* and *Streptomyces ghanaensis*.

††All feed products with this AAI had the same strength.

common AAIs listed were enramycin (18.8% chicken feeds), followed by bacitracin (16.5%), chlortetracycline (15.3%), avilamycin (5.9%), flavomycin (4.6%), colistin (3.7%), virginamycin (2.4%), and oxytetracycline (1.2%) (Table 1). A total of 10 (9.3%) products were not compliant with Regulation 06/2016/TT-BNNPTNT, either because they included a non-authorised AAI (avilamycin, flavomycin, oxytetracycline) (n = 4), AAI/s over the permitted limits (n = 4), or for both reasons (n = 2).

## AMU through commercial feed intake

All flocks were raised on commercial chicken feed. In addition, pig and duck feeds were given to 12.1% and 0.6% flocks, respectively. Detailed information about the feed and AAI consumption in each flock is provided in the S1 Data. Each flock had been given a median of 2 [Interquartile range (IQR) 2–3] different commercial feed products. Flocks received a median of 1 [IQR 1–1] antimicrobial-containing products. Overall, the highest amounts of AMU corresponded to chlortetracycline (42.2mg, SEM±0.34), followed by enramycin (18.4mg, SEM ±0.03) and bacitracin (16.4mg SEM ±0.20) (Table 2).

**Table 2. AMU in commercial feed among 338 small-scale chicken flocks over 6,041 observation weeks.**

| AAIs | No. flocks(n = 338) (%) | Prevalence of AMU by week (mean ± SEM) [lowest-highest] | Total AMU over the production cycle mg/kg chicken (mean ± SEM) [lowest-highest] (%) |
|---|---|---|---|
| Enramycin | 152 (45.4) | 0.319 (± 0.004) [0.306–0.333] | 18.4 (± 0.03) [17.3–19.5] (21.8) |
| Chlortetracycline | 73 (22.5) | 0.134 (± 0.002) [0.134–0.135] | 42.2 (± 0.34) [40.6–43.9] (50.0) |
| Bacitracin | 103 (30.5) | 0.095 (± 0.014) [0.080–0.111] | 16.4 (± 0.20) [10.5–22.3] (19.4) |
| Virginamycin | 8 (2.9) | 0.010 (± 0.032) [0.005–0.014] | 0.50 (± 0.17) [0.1–0.8] (0.6) |
| Colistin* | 7 (2.0) | 0.005 (± 0.037) [0.003–0.008] | 6.40 (± 4.21) [2.6–10.3] (7.6) |
| Avilamycin | 8 (2.3) | 0.005 (± NC) [0.0–0.010] | 0.30 (± 0.08) [0.0–0.6] (0.4) |
| Flavomycin | 8 (2.3) | 0.005 (± NC) [0.0–0.010] | 0.20 (± 0.11) [0.0–0.4] (0.2) |
| Oxytetracyline | 4 (1.1) | 0.0 (± NC) [0.0–0.001] | 0.07 (± 0.73) [0.0–0.15] (0.1) |

NC = Not calculated.

*Critically-important antimicrobial class according to the World Health Organization.

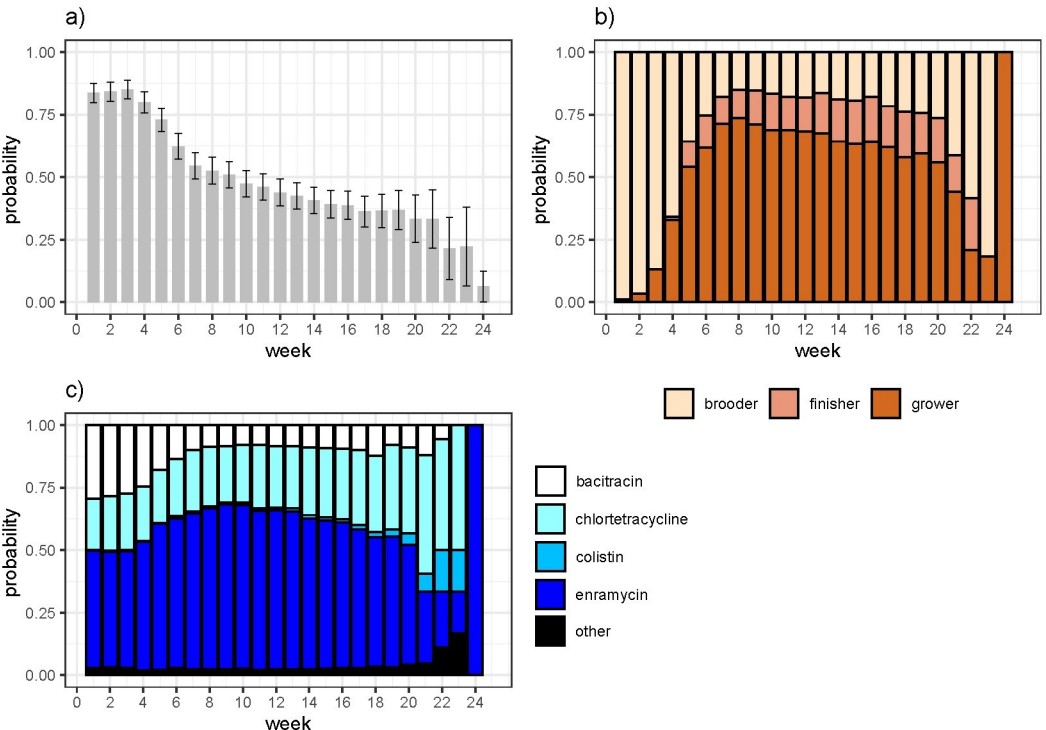

**Fig 1.** (a) Probability of consumption of AAIs in chicken feeds by week among study flocks; (b) Weekly distribution of types of feed (production stage) consumed by flocks; (c) Weekly distribution of AAIs consumed by flocks through commercial animal feeds.

Commercial feed rations were given to flocks over a total of 5,655 of 6,041 (93.6%) observation weeks. The probability of AMU in flocks decreased with the age of the flock (Fig 1A). On average, flocks were given AGPs in feed on 57.5% (SEM ±2.8%) weeks. Interestingly, a relatively high fraction of brooder products were used in later stages, while some finisher products were also used more in the growing period. Enramycin was used predominantly throughout the production cycle, while colistin was found only in later stages (Fig 1C).

## Discussion

There are very few published data describing and quantifying consumption of AAIs in commercial feeds in poultry farming systems in LMICs [25]. Our findings complement existing data on antimicrobials administered (mainly through water) to Mekong Delta flocks [19]. Quantitatively, consumption of in-feed antimicrobials represents a small fraction (~10%) of total AMU, a figure consistent with previous estimates [16, 17].

This study is based on data from a large cohort study aiming at reducing AMU in chicken production in the Mekong Delta of Vietnam [20]; therefore we believe that results are representative of commercially chicken farming systems, since our farm selection was random. Even though our data came from an intervention study, our advice to farmers focused on reducing prophylactic and therapeutic administration of antimicrobials in water, and did not include advice on feed. Furthermore, we did not find any difference in in-feed antimicrobial consumption between flocks allocated to the intervention compared with the baseline phase (data not shown).

A major concern is the relatively high number of products that did not comply with Vietnamese regulations. Bacitracin, banned in feed rations in Vietnam since May 2016 [23], was

the second most common AAI. More worryingly, we found that 17% (6/35) antimicrobial-containing feeds included AAIs at concentrations higher than permitted by Vietnamese authorities. Notably, the strength of colistin was 3–5 times greater than permitted in all products examined, and non-authorised antimicrobials (avilamycin, flavomycin, oxytetracyline) were found in some chicken feeds. This raises concerns regarding compliance of commercial feed mills with regulations, and casts doubts over the effective implementation of the phased bans [18]. An additional challenge is the ambiguous labelling of their AAI content in about a third of the rations investigated.

Recent studies have reported a high prevalence of colistin resistance encoded by *mcr-1* in commercial chicken flocks in the area [17]. This antimicrobial, classified as highly critically important by WHO [26], was listed in 5% of feeds examined (brooder feeds) and we estimated that, on average, flocks consumed 6.40 mg/kg of this antimicrobial (about 3% of total in-feed AMU). This is a modest amount compared with the reported magnitude of AMU through water administration (42.0 mg/kg). However, it is of concern that in our study farms these medicated feeds were predominantly administerd towards the end of the production cycle, and may pose a risk of antimicrobial residues in poultry meat [27]. It is of great concern that only 16/35 (45.7%) AGP-containing feeds examined did not mention withdrawal times (data not shown). A recent survey showed that 8.4% of chicken meat samples in Vietnam contained antimicrobials residues, tetracyclines being the most common residue detected [28].

Quantitatively, in our study chlortetracycline, bacitracin and enramycin were the AAIs most consumed through commercial feeds. These results are not dissimilar to previous extrapolations from a retail survey in Vietnam [16]. Tetracyclines were also the most consumed antimicrobials consumed by flocks through water [19], and the antimicrobial class against which resistance among *Escherichia coli* and non-typhoidal *Salmonella* strains in the Mekong Delta is highest [17, 29, 30]. Bacitracin use has been shown to promote resistance among *Clostridium perfringens* isolates from chickens [31, 32]. With regards to enramycin, there is little information on its impact on AMR. A Japanese study that investigated *Enterococcus faecium* isolates from chicken flocks found no evidence of resistance against enramycin, although the study presented no enramycin use data [33].

Much of the debate on AMU in animals has often been framed in terms of bans on AGPs. Unfortunately, global data on total amounts of AGPs consumed or on the contribution of AGPs on total AMU are lacking. In Great Britain, in 2001 (5 years before the 2006 EU ban), AGPs represented 11.6% of 371 tonnes of antimicrobial active ingredients used in animal production [34]. We believe that AGPs represent a considerable fraction of total AMU globally, although probably these quantities have been decreasing over recent years, since more and more countries have phased out their use. A recent OIE survey reports that AGPs were used in 23% countries surveyed in 2018, compared with 51% countries in 2012 [35]. A review of the data of the impact of AGP from 1950 to 2010 on farm productivity indicate that gains due to AGP in feeds have been decreasing over the years [36], and suggest that their effects are of greater magnitude in low-biosecurity systems. Indeed, recent studies in industrial (short cycle) broiler production systems showed that AGPs added to feed did not overall improve flock bodyweight [37, 38]. In the non-industrial production systems investigated here, antimicrobials in commercial feeds represented a relatively small fraction of total AMU. It is conceivable that even if AGPs may have led to marginal productivity gains, these are likely to have been offset by the high mortality rates due to pathogen circulation common in the area [39].

This study provides preliminary quantification on AGP consumption in native chicken production in the country based on examination of feed products' labels. However, this information should be taken with caution since a certain degree of inaccuracy in the AAI strength in feed labels is expected. Furthermore, more data are needed in order to accurately establish

quantify levels of AGPs (and total AMU) in pig production, which by far for the most commonly consumed type of meat in Vietnam. Data from a survey in Vietnam suggest that in-feed antimicrobial consumpton in pigs is of greater magnitude than in chicken production [16]. Similar to Vietnam, the use of medicated feed in pig production in Thailand is common practice [40]. Because of a higher magnitude of use, the impact of reductions or bans on AGPs in the pig species is uncertain.

## Conclusions

Compared with antimicrobials administered through water, antimicrobials in feed represent a relatively small fraction (10%) of total AMU in Vietnamese chicken production. However, it is of great concern that some feed formulations examined included colistin (polymyxin E), a critically important antimicrobial of highest priority for human medicine. Furthermore, a considerable number of feed formulations did not comply with Vietnamese government regulations with regards to their AAI content, strength and/or withdrawal times, suggesting that effective enforcing and monitoring of such restrictions in the country may be challenging. The types and amounts of antimicrobials in feeds will necessarily differ from country to country as well as by production system; therefore more data on AGPs in feed are needed in order to formulate targeted policy initiatives that are appropriate for specific situations.

## Supporting information

**S1 Table. Detailed information on antimicrobial-containing feed formulations intended for chickens.**
(DOCX)

**S1 Data. Weekly estimated consumption data of feed and AAIs contained in feed products of study flocks.**
(XLSX)

## Acknowledgments

We are grateful to all participating farmers and field staff. We thank staff at SDAH-DT for logistic support.

## Author Contributions

**Conceptualization:** Nguyen Van Cuong, Vo Be Hien, Juan Carrique-Mas.

**Data curation:** Bach Tuan Kiet, Doan Hoang Phu, Juan Carrique-Mas.

**Formal analysis:** Nguyen Van Cuong, Bach Tuan Kiet, Doan Hoang Phu, Marc Choisy, Juan Carrique-Mas.

**Investigation:** Doan Hoang Phu, Marc Choisy.

**Methodology:** Vo Be Hien, Doan Hoang Phu, Marc Choisy.

**Project administration:** Vo Be Hien, Juan Carrique-Mas.

**Resources:** Juan Carrique-Mas.

**Supervision:** Juan Carrique-Mas.

**Validation:** Nguyen Van Cuong, Marc Choisy.

**Visualization:** Guy Thwaites.

**Writing – original draft:** Bao Dinh Truong, Guy Thwaites, Juan Carrique-Mas.

**Writing – review & editing:** Nguyen Van Cuong, Bao Dinh Truong, Guy Thwaites, Juan Car-
rique-Mas.

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
