## [Decision Letter · Decision Letter 0]

22 Jan 2021

PONE-D-20-36867

Antimicrobial use through consumption of medicated feeds in chicken flocks in the Mekong Delta of Vietnam: a three-year study before a ban on antimicrobial growth promoters

PLOS ONE

Dear Dr. Carrique-Mas,

Thank you for submitting your manuscript to PLOS ONE. After careful consideration, we feel that it has merit but does not fully meet PLOS ONE’s publication criteria as it currently stands. Therefore, we invite you to submit a revised version of the manuscript that addresses the points raised during the review process.

Explanations and clarifications needed for number of points raised by the reviewers

We look forward to receiving your revised manuscript.

Kind regards,

Iddya Karunasagar

Academic Editor

PLOS ONE

Journal Requirements:

4.Your ethics statement should only appear in the Methods section of your manuscript. If your ethics statement is written in any section besides the Methods, please move it to the Methods section and delete it from any other section. Please ensure that your ethics statement is included in your manuscript, as the ethics statement entered into the online submission form will not be published alongside your manuscript.

Additional Editor Comments:

Reviewers have pointed out number of issues in the manuscript. The authors seem to have included data published by them earlier in this manuscript. There are number of other clarification and explanations required. Please address all reviewer comments point by point.

Reviewers' comments:

Reviewer's Responses to Questions

**Comments to the Author**

1. Is the manuscript technically sound, and do the data support the conclusions?

Reviewer #1: Partly

Reviewer #2: Yes

Reviewer #3: Yes

Reviewer #4: Yes

2. Has the statistical analysis been performed appropriately and rigorously? 

Reviewer #1: N/A

Reviewer #2: I Don't Know

Reviewer #3: Yes

Reviewer #4: Yes

3. Have the authors made all data underlying the findings in their manuscript fully available?

Reviewer #1: Yes

Reviewer #2: Yes

Reviewer #3: Yes

Reviewer #4: Yes

4. Is the manuscript presented in an intelligible fashion and written in standard English?

Reviewer #1: Yes

Reviewer #2: Yes

Reviewer #3: Yes

Reviewer #4: No

5. Review Comments to the Author

Reviewer #1: The manuscript described the amounts of antimicrobials in commercial feed products given to chicken flocks in the Mekong Delta of Vietnam. Several feed formulations included colistin and non-authorized antimicrobials. The authors highlighted the challenges for restriction of antimicrobials in commercial feeds in low-and middle-income countries.

1) Although results seem to be representative of commercially chicken farming systems in the Mekong Delta of Vietnam because the study is based on a large cohort study and farms participated were randomly selected, this does not mean that the interpretation can generalize down to low-and middle-income countries. The data were obtained from information in labels of feed products consumed in the Mekong Delta of Vietnam. Details of consumption of commercial feeds in the other countries were not investigated in this study. The sentence described in lines 99-102 should be deleted because of the lack of evidences.

2) Logic in the discussion section is sometimes hardly understandable.

- Descriptions in lines 248-254 should be deleted because the topic is beyond the scope of this study as the authors mentioned in line 254.

- The context in lines 255-258 is associated with that in the sentence starting in line 235

- In lines 263-280, the point is unclear and the story developed in unexpected directions.

3) Several references are not properly formatted.

4) Supporting information (S1 Table. This is the S1 Table Title. This is the S1 Table legend.) is not included in the manuscript, which may be situated after the reference section.

5) Minor points

- “One tenth of the formulations examined did not comply with Government regulations” (lines 37-38) is a repetition of the sentence “A total of 10 (9.3%) products were not compliant with existing Vietnamese regulation --- ” (lines 32-33).

- line 85: “olaquidox” should be replaced with “olaquindox.”

- line 110: “all-in all-out” or “all-in/all-out” are usually used.

- “We excluded ionophores since --- ” (lines 129-132) is a repetition of the sentence “Ionophores (mostly aimed at controlling coccidial infection) were excluded” (lines 124-125).

- line 166: “finisher (12.5%)” should be replaced with “finisher (14.3%)” because the number of products consumed during the finishing stage seems to be 5, thus 5/35=14.3%, according to Supplementary Table.

Reviewer #2: The authors present a study of antimicrobial use in commercial feed in a rather large representative cohort in small-scale chicken farms in Vietnam. This study provides information on the types and quantities of the antimicrobials consumed in small-scale commercial chicken farms in the area.

Although the manuscript provides quantitative profiles of the antimicrobial active ingredients (AAIs) in the commercial feed products, there's no available information on antimicrobial resistance rates in major bacterial parhogens such as E. coli, Salmonella, Enterococcus spp., or S. aureus etc. isolated from the study sites.

As presented the manuscript is not acceptable for publication in PLOS One as a full-length article.

Reviewer #3: This is an interesting paper on a very relevant subject. The paper is well written and the results are clearly presented and discussed.

I have only a limited number of minor remarks.

Line 29: please add unit

Line 46: per year?

Line 50-51: do you refer to AMR in general or AMR in humans? I think this sentence should be specified a bit more as it can be misinterpreted. As far as the AMU in animals contributes to the AMR in humans it is doubtful whether this can be categorized as “major contribution”?

Line 51-53: do you believe this statement yourself? Given the fact that there are many legislation changes with regard to use of growth promotors. I think you should add to the sentence “if nothing changes with regard to AMU”

Line 64: something wrong in this sentence

Line 69: some countries outside the EU

Line 78: “commercial” ?? should this not be “antibiotics”?

Line 260-262: What is not entirely clear here?

Line 279-280: I dot fully understand the link with Thailand?

Reviewer #4: This paper explores the use of antimicrobials in meat chicken production in a specific area of Vietnam. The authors have collected a great dataset with which to estimate AMU during the growout cycle of the flock. A major concern I have is the similarity of this paper to the numerous others published by this research group using the same dataset. Within the reference section alone, there are several that seem to have the same AGP data within (see references 16, 17, 19, 26, 44. Note that references 19 and 26 are the same.). The paper does provide a view on AMU in an LMIC environment.

The paper has many errors in grammar and syntax. Please proof the entire manuscript.

Other concerns:

Line 48: Is the statement “all of which have well-developed antimicrobial consumption surveillance” actually correct? It is my understanding that these surveillance programs are based on sales data. Only a few countries have actual surveillance of on-farm use. Sales data are not the same as usage data.

Line 55: Why over the last 5 years? This issue has been discussed and debated for decades. Also, the issue is not solely related to “excessive” use but any antimicrobial use, as any use at any location can potentially select for resistant bacterial populations.

Line 64: Delete “are” before “often”

Line 75: “stoods”?

Line 78: “commercial are”?

Lines 100-103: How will the knowledge of data on in-feed consumption in chicken farming for the basis of informed decisions? If the data only include the total amounts of drug used without context of the diseases that they are preventing, controlling or treating, then how will these data be used?

Line 112: How is “reducing AMU in chicken production” an aim of a longitudinal study. This seems to assume, a priori, that reduction is needed. Shouldn’t the appropriate objective of a longitudinal study be to evaluate the ways in which antimicrobials are being used and help producers and veterinarians ensure that antimicrobials are being used responsibly? This is not the same as reducing use. This potential bias of a goal is stated again on line 212.

Lines 120 and 142: The study appears to be focused on meat chickens, but line 120 says that the duration of a production cycle is 19 weeks. Is this correct? Line 140 says that feed consumption was estimated from data on layer pullets, but a layer chicken is very different from a meat chicken. Were these data validated?

Table 1 has a strange format due to the way in which the ranges are presented. If there is no range (because the drug is only used at a single concentration) then the table should probably state this rather than showing the same number twice.

Table 1: Flavomycin is in the bambermycins class.

Lines 186-187: Move the unit “AAI/kg” after the number 84.8.

Line 190: the use of the term “magnitude” is based on what? Total amount of drug? The paper does not seem to adjust for potency and thus magnitude can be a misleading term if not defined.

Table 2: What is a mean probability? How was this estimated? And are these really probabilities? Use is not a random variable, so is this really just a prevalence of use by week?

Line 198: Why are the authors using 95% CI for these prevalence estimates. They know the exact number. If they are extrapolating to the larger chicken industry, then perhaps the CI is warranted, but here shouldn’t the authors report the standard error (or deviation, depending on the data that were used to generate the point estimate)?

Lines 236-237: If there is the potential for residues in the meat, then the authors need to explain in more detail the withdrawal times that are approved for different drugs in Vietnam. This is not an issue in many countries as there are rules around the amount of time that specific drugs need to be out of the feed prior to slaughter.

Lines 249-252: This statement has now been shown to be entirely false. There are many scientific papers and books that have shown that VRE continued to increase in human patients in Europe even after the ban of avoparcin, and other countries which never approved avoparcin had even higher VRE levels then some of the European countries. It is not clear why this example was given, as it is not relevant to the study.

Lines 259-262: This paragraph seems to make a strongly opinionated statement, but due to poor English grammar, the meaning is not understandable to this reviewer.

Lines 272-273: This sentence absolutely must be eliminated or modified. The studies did not show that at all.

Lines 275-276: This statement is unclear, as the authors own study reported in the abstract that “In flocks reporting disease, AMU significantly reduced the incidence of mortality (HR=0.90).” So it would appear that although overall disease incidence and mortality are high, AMU helps control the impact of the disease. Again, it appears that the paper is written with an agenda, where any AMU is bad and that the goal should be to reduce use regardless of whether disease is being treated and controlled with the AMU.

6. PLOS authors have the option to publish the peer review history of their article (what does this mean?). If published, this will include your full peer review and any attached files.

Reviewer #1: **Yes: **Toshiyuki Murase

Reviewer #2: No

Reviewer #3: **Yes: **Jeroen Dewulf

Reviewer #4: No

---

## [Author Response · Author response to Decision Letter 0]

28 Jan 2021

Reviewer #1: The manuscript described the amounts of antimicrobials in commercial feed products given to chicken flocks in the Mekong Delta of Vietnam. Several feed formulations included colistin and non-authorized antimicrobials. The authors highlighted the challenges for restriction of antimicrobials in commercial feeds in low-and middle-income countries.

1) Although results seem to be representative of commercially chicken farming systems in the Mekong Delta of Vietnam because the study is based on a large cohort study and farms participated were randomly selected, this does not mean that the interpretation can generalize down to low-and middle-income countries. The data were obtained from information in labels of feed products consumed in the Mekong Delta of Vietnam. Details of consumption of commercial feeds in the other countries were not investigated in this study. The sentence described in lines 99-102 should be deleted because of the lack of evidences.

Response - We understand and agree with the reviewer the fact that the situation described here can be extrapolated to other countries is a strong assumption. This is indeed based on unconfirmed anecdotal evidence (after having talked to other colleagues in conferences, etc.). We have now removed the sentence as required. 

2) Logic in the discussion section is sometimes hardly understandable.

Response - We have extensively checked the manuscript and improved the grammar in several places, including the Discussion section.

- Descriptions in lines 248-254 should be deleted because the topic is beyond the scope of this study as the authors mentioned in line 254.

Respose - This has been deleted as suggested.

- The context in lines 255-258 is associated with that in the sentence starting in line 235.

Respose - This has now been arranged. We have moved forward the paragraph from 255-258 to appear after line 235. 

- In lines 263-280, the point is unclear and the story developed in unexpected directions. 

Response - We wanted to make the point that there are limited data on how much of the total AMU in animal production is in the form of AGPs. We have slightly reworded this sentence as follows: “There are very limited data on the contribution of AGPs in relation on total AMU in animal production. In Great Britain, in 2001 (5 years before the 2006 EU ban), AGPs represented 11.6% of 371 tonnes of antimicrobial active ingredients used in animal production [34]. We believe that AGPs represent a considerable fraction of total AMU globally, and probably these quantities have been decreasing over recent years, since more and more countries have phased out their use.”

3) Several references are not properly formatted.

Response - The references have been carefully checked to comply with the journal’s requirement.

4) Supporting information (S1 Table. This is the S1 Table Title. This is the S1 Table legend.) is not included in the manuscript, which may be situated after the reference section.

Response - We have now re-labelled this supplementary material as ‘S1 Table’, and included a caption. The data is available as an Excel sheet file (S2_Data).

5) Minor points

- “One tenth of the formulations examined did not comply with Government regulations” (lines 37-38) is a repetition of the sentence “A total of 10 (9.3%) products were not compliant with existing Vietnamese regulation --- ” (lines 32-33).

Response - This has now been amended.

- line 85: “olaquidox” should be replaced with “olaquindox.”

Response - This has now been amended.

- line 110: “all-in all-out” or “all-in/all-out” are usually used.

Response – This has now been amended.

- “We excluded ionophores since --- ” (lines 129-132) is a repetition of the sentence “Ionophores (mostly aimed at controlling coccidial infection) were excluded” (lines 124-125).

Response - This has now been amended, and removed the second statement in relation to ionophores. 

- line 166: “finisher (12.5%)” should be replaced with “finisher (14.3%)” because the number of products consumed during the finishing stage seems to be 5, thus 5/35=14.3%, according to Supplementary Table.

Response - This was a mistake. We have now chosen to relate these percentages in relation to feed products intended for chickens, not all feed products (a number of flocks consumed feed formulations intended for pigs and ducks). These were 38 intended for brooding, 22 for growers (i.e mid-production) and 25 for finishing. The number (and %) of feed products with antimicrobials was 25 (65.8%), 9 (40.9%) and 5 (20%), respectively. We have amended the text now. 

 

Reviewer #2: The authors present a study of antimicrobial use in commercial feed in a rather large representative cohort in small-scale chicken farms in Vietnam. This study provides information on the types and quantities of the antimicrobials consumed in small-scale commercial chicken farms in the area.

Although the manuscript provides quantitative profiles of the antimicrobial active ingredients (AAIs) in the commercial feed products, there's no available information on antimicrobial resistance rates in major bacterial parhogens such as E. coli, Salmonella, Enterococcus spp., or S. aureus etc. isolated from the study sites.

As presented the manuscript is not acceptable for publication in PLOS One as a full-length article. 

Response - The topic of this study is limited to AGPs in feed. The investigation of resistance in flocks is beyond the scope of the study. 

 

Reviewer #3: This is an interesting paper on a very relevant subject. The paper is well written and the results are clearly presented and discussed.

I have only a limited number of minor remarks.

Line 29: please add unit.

Response - This should be ‘mg’. Now amended. 

Line 46: per year?

Response - The sentence mentions ‘annual’ already.

Line 50-51: do you refer to AMR in general or AMR in humans? I think this sentence should be specified a bit more as it can be misinterpreted. As far as the AMU in animals contributes to the AMR in humans it is doubtful whether this can be categorized as “major contribution”?

Response - We have re-worded this sentence to qualify this link. ‘It is believed that excessive use of antimicrobials in animal production is one factor contributing to the global rise in antimicrobial resistance (AMR), although the magnitude of it is unknown [3, 4]’.

Line 51-53: do you believe this statement yourself? Given the fact that there are many legislation changes with regard to use of growth promotors. I think you should add to the sentence “if nothing changes with regard to AMU”.

Response - I do believe consumption of antimicrobials intended for animal production will generally increase in coming years, mostly driven by increased in production associated with increased demand of animal protein in many LMICs. 

Line 64: something wrong in this sentence.

Response - The grammar in this sentence was not correct. We have now amended it (i.e. removing the word ‘are’).

Line 69: some countries outside the EU.

Response - This has added now. 

Line 78: “commercial” ?? should this not be “antibiotics”?

Response - Yes, this was a mistake. We have now added ‘antimicrobials’. 

Line 260-262: What is not entirely clear here?

Response - This was a mistake. The whole paragraph has now been amended: ‘Much of the debate on AMU in animals has often been framed in terms of bans on AGPs. Unfortunately, accurate data on total amounts of AGPs consumed or on the contribution of AGPs on total AMU. In Great Britain, in 2001 (5 years before the 2006 EU ban), AGPs represented 11.6% of 371 tonnes of antimicrobial active ingredients used in animal production [34]. We believe that AGPs represent a considerable fraction of total AMU globally, and probably these quantities have been decreasing over recent years, since more and more countries have phased out their use’. 

Line 279-280: I dot fully understand the link with Thailand?

Response - Thailand is a country that shares may features with Vietnam (located in SE Asia, both are LMICs, etc), although we acknowledge that the original paragraph mentioned by the reviewer is written confusingly. We have now amended it to better express the idea that in the Southeast Asian region, AGP use in pig production appears to be of greater magnitude than in chicken production.

 

Reviewer #4: This paper explores the use of antimicrobials in meat chicken production in a specific area of Vietnam. The authors have collected a great dataset with which to estimate AMU during the growout cycle of the flock. A major concern I have is the similarity of this paper to the numerous others published by this research group using the same dataset. Within the reference section alone, there are several that seem to have the same AGP data within (see references 16, 17, 19, 26, 44. Note that references 19 and 26 are the same.). The paper does provide a view on AMU in an LMIC environment.

Response - Yes, references 19 and 26 are the same study. This has now been amended. We acknowledge that the reference 44 mentioned here addresses a subset of the data from the same cohort, but in that study there consumption of AGPs in feed was not measured. References 16 and 17 correspond to completely different studies (and datasets). Reference 16 is a study based on a retail study of commercial feeds available (by internet) in the country, and reference 17 is a separate study of only 12 farms (in a different province).

The paper has many errors in grammar and syntax. Please proof the entire manuscript.

Reference - The manuscript has been thoroughly checked and a number of mistakes spotted have been amended.

Other concerns:

Line 48: Is the statement “all of which have well-developed antimicrobial consumption surveillance” actually correct? It is my understanding that these surveillance programs are based on sales data. Only a few countries have actual surveillance of on-farm use. Sales data are not the same as usage data.

Response - Sensu stricto the reviewer is correct, sales does exactly mean consumption. However, in practical terms, the consumption is estimated from sales data. Even the title of the ESVAC reports use the term “consumption”. 

Line 55: Why over the last 5 years? This issue has been discussed and debated for decades. Also, the issue is not solely related to “excessive” use but any antimicrobial use, as any use at any location can potentially select for resistant bacterial populations.

Response - Yes, it is true that this comes from the last two decades, only that since 2015 there has been much notably more pressing interest. We have changed this sentence to ‘over the last years’.

Line 64: Delete “are” before “often”.

Response - This has been now amended. 

Line 75: “stoods”?

Response - This has now been amended.

Line 78: “commercial are”?

Response – This has been corrected. “Commercial” should be “antimicrobial”.

Lines 100-103: How will the knowledge of data on in-feed consumption in chicken farming for the basis of informed decisions? If the data only include the total amounts of drug used without context of the diseases that they are preventing, controlling or treating, then how will these data be used?

Response - This would require additional research that this manuscript does not address. The factual knowledge on AMU through AGP consumption, is however important knowledge that should be used as a benchmark for future reductions. We have amended the text to qualify this statement. 

Line 112: How is “reducing AMU in chicken production” an aim of a longitudinal study. This seems to assume, a priori, that reduction is needed. Shouldn’t the appropriate objective of a longitudinal study be to evaluate the ways in which antimicrobials are being used and help producers and veterinarians ensure that antimicrobials are being used responsibly? This is not the same as reducing use. This potential bias of a goal is stated again on line 212.

Response - This was probably not clearly expressed here. What we mean is that this study was performed in the framework of a large intervention study (trial) using a longitudinal study design. The reference given (Carrique-Mas and Rushton, 2017) refers to that study. We have re-worded this now: ‘This study was performed in the context of a large field based trial aimed at reducing AMU in chicken production through the provision of veterinary advice [20].’

Lines 120 and 142: The study appears to be focused on meat chickens, but line 120 says that the duration of a production cycle is 19 weeks. Is this correct?

Response - Yes, these are slow growing, native breeds that take longer to grow than a typical industrial broiler chicken. This has now been clarified in the text. 

Line 140 says that feed consumption was estimated from data on layer pullets, but a layer chicken is very different from a meat chicken. Were these data validated?

Response – The type of native meat chicken investigated here is very similar in terms of physiology and growth to layer pullets. Therefore we believe it is reasonable to use consumption data typical of layer breeds, and not of industrial broilers.

Table 1 has a strange format due to the way in which the ranges are presented. If there is no range (because the drug is only used at a single concentration) then the table should probably state this rather than showing the same number twice.

Response – This has been now done. 

Table 1: Flavomycin is in the bambermycins class.

Response – Flavomycin is synonymous to ‘bambermycin’, which is in itself a complex antimicrobial substance.

Lines 186-187: Move the unit “AAI/kg” after the number 84.8.

Response – This has been amended now.

Line 190: the use of the term “magnitude” is based on what? Total amount of drug? The paper does not seem to adjust for potency and thus magnitude can be a misleading term if not defined.

Response – We do not fully understand this comment. In this study we were able to quantify amunts of AGPs consumed based on estimated amounts of feed consumed and the strength of the AAI in the feed product. However we understand that it can be misleading. We replaced this term by ‘amounts of AMU’.

Table 2: What is a mean probability? How was this estimated? And are these really probabilities? Use is not a random variable, so is this really just a prevalence of use by week?

Response – We agree with this comment. Accordingly, we have now amended the heading of the corresponding column to rename it: ‘Prevalence of AMU by week’. 

Line 198: Why are the authors using 95% CI for these prevalence estimates. They know the exact number. If they are extrapolating to the larger chicken industry, then perhaps the CI is warranted, but here shouldn’t the authors report the standard error (or deviation, depending on the data that were used to generate the point estimate)?

Response – We have replaced this now by the mean (57.5%) and the Standard Error of the Mean (SEM ±2.8% weeks).

Lines 236-237: If there is the potential for residues in the meat, then the authors need to explain in more detail the withdrawal times that are approved for different drugs in Vietnam. This is not an issue in many countries as there are rules around the amount of time that specific drugs need to be out of the feed prior to slaughter.

Response – The current Vietnamese legislation (Decision 39/2017/NĐ-CP) states that the products need to specify withdrawal times (based on the technical dossier required by each product), but does not list specific AGPs and their associated withdrawal times. We found in our study that 16 of the 35 products did not mention withdrawal times, therefore representing a further 

Lines 249-252: This statement has now been shown to be entirely false. There are many scientific papers and books that have shown that VRE continued to increase in human patients in Europe even after the ban of avoparcin, and other countries which never approved avoparcin had even higher VRE levels then some of the European countries. It is not clear why this example was given, as it is not relevant to the study.

Response- This paragraph has been deleted altogether.

Lines 259-262: This paragraph seems to make a strongly opinionated statement, but due to poor English grammar, the meaning is not understandable to this reviewer.

Response –Our intention was to make the point that the amounts of AGP consumption in relation to total AMU are largely unknown. The exception is some limited data from the UK. We have now reworded this paragraph: ‘Much of the debate on AMU in animals has often been framed in terms of bans on AGPs. Unfortunately, global data on total amounts of AGPs consumed or on the contribution of AGPs on total AMUare lacking. In Great Britain, in 2001 (5 years before implementation of the 2006 EU ban), AGPs represented 11.6% of 371 tonnes of antimicrobial active ingredients used in animal production [34]’.

Lines 272-273: This sentence absolutely must be eliminated or modified. The studies did not show that at all.’

Response - The study by Kumar et al. 2018 is based on an experimental randomised study on 240 chickens. In terms of bodyweight, the study showed better performance at the end of the 42-day production cycle, among birds not receiving supplementation with bacitracin methylene disalicylate (BMD) compared with those receiving this supplement. Another study by Hamid et al (2) carried out an experiment using four types of AGPs and did not show overall differences in growth over the whole production period (0-49 days). We have qualified the statement in the text in the view of these other published studies. 

Lines 275-276: This statement is unclear, as the authors own study reported in the abstract that “In flocks reporting disease, AMU significantly reduced the incidence of mortality (HR=0.90).” So it would appear that although overall disease incidence and mortality are high, AMU helps control the impact of the disease. Again, it appears that the paper is written with an agenda, where any AMU is bad and that the goal should be to reduce use regardless of whether disease is being treated and controlled with the AMU.

Response – We presume that the reviewer refers here to the previous study https://doi.org/10.1016/j.prevetmed.2019.02.005 (performed using a subset of farms included in the current study on AGPs). In that study the use of oral antimicrobials in water demonstrated a reduction in mortality when used in situations of disease (AGPs were not investigated in that study). The types of antimicrobials administered orally in water are very different to those in feed as AGPs. They are typically broad spectrum and many are CIAs. Also the strenghts and total amounts consumed in water are considerably higher. That study did not concern the use of in-feed AGPs. We absolutely have no agenda with regards to AGP. We just mention that they may or may not help control disease

---

## [Decision Letter · Decision Letter 1]

18 Mar 2021

PONE-D-20-36867R1

Antimicrobial use through consumption of medicated feeds in chicken flocks in the Mekong Delta of Vietnam: a three-year study before a ban on antimicrobial growth promoters

PLOS ONE

Dear Dr. Carrique-Mas,

Thank you for submitting your manuscript to PLOS ONE. After careful consideration, we feel that it has merit but does not fully meet PLOS ONE’s publication criteria as it currently stands. Therefore, we invite you to submit a revised version of the manuscript that addresses the points raised during the review process.

Please address some comments missed in the last revision 

We look forward to receiving your revised manuscript.

Kind regards,

Iddya Karunasagar

Academic Editor

PLOS ONE

Journal Requirements:

Additional Editor Comments (if provided):

Some of the previous comments of the reviewers have not been addressed. Please look into the comments and revise the manuscript.

Reviewers' comments:

Reviewer's Responses to Questions

**Comments to the Author**

1. If the authors have adequately addressed your comments raised in a previous round of review and you feel that this manuscript is now acceptable for publication, you may indicate that here to bypass the “Comments to the Author” section, enter your conflict of interest statement in the “Confidential to Editor” section, and submit your "Accept" recommendation.

Reviewer #1: (No Response)

Reviewer #3: All comments have been addressed

Reviewer #4: (No Response)

2. Is the manuscript technically sound, and do the data support the conclusions?

Reviewer #1: Yes

Reviewer #3: Yes

Reviewer #4: Yes

3. Has the statistical analysis been performed appropriately and rigorously? 

Reviewer #1: N/A

Reviewer #3: Yes

Reviewer #4: Yes

4. Have the authors made all data underlying the findings in their manuscript fully available?

Reviewer #1: Yes

Reviewer #3: Yes

Reviewer #4: Yes

5. Is the manuscript presented in an intelligible fashion and written in standard English?

Reviewer #1: Yes

Reviewer #3: Yes

Reviewer #4: No

6. Review Comments to the Author

Reviewer #1: lines 252-254: According to the authors, they have slightly reworded this sentence as follows: “There are very limited data on the contribution of AGPs in relation on total AMU in animal production. In Great Britain,---” although the first sentence is not identical to the revised manuscript. Should this part read “Much of the debate on AMU in animals has often been framed in terms of bans on AGPs. Unfortunately, global data on total amounts of AGPs consumed or on the contribution of AGPs on total AMU are lacking” in lines 252-254 of the revised MS? If so, this reviewer has no additional comment.

lines 163-167: If the number of antimicrobial-containing feed products intended for chickens is thirty-five (line 163), description about the breakdown (lines 164-166) is incorrect. According to the S1 Table, twenty-one (60%, 21/35), 9 (25.7%, 9/35), and 5 (14.3%, 5/35) products are for brooding, growing, and finishing, respectively.

S1 Table1: The title should include the term “chicken.” For example, Detailed information on antimicrobial-containing feed products intended for chickens.

Reviewer #3: The authors have adequately answered to all my comments, i have no further comments therefore i suggest this paper to be accepted for publication

Reviewer #4: The paper still has errors in grammar and syntax. This is unfortunate because PLoS does no copyediting, one reason not to publish in PLoS. Please proof the entire manuscript.

Remaining concerns:

Line 48: The authors did not address my concern regarding the statement that all EU countries “have well-developed antimicrobial consumption surveillance.” There is nothing well-developed about an AMU surveillance system that is completely based on sales data. It does not matter that ESVAC uses the term “consumption” in the title of the report. The fact is that the EU surveillance is based on sales data, and the countries do not have any information that would be important for characterizing AMU in animals. Most importantly, there are no data regarding the indication for use (disease). Of course, there also no data on dose, duration, number of animals exposed, etc. The ESVAC reports briefly mention these pitfalls. While AMU surveillance using sales data is an easy target for many countries, it should not be considered well-developed, complete, ideal or even useful for assessing whether uses are responsible and whether stewardship principles are being followed.

Line 50: Define AMU here

Line 58: Remove AMU definition

Line 70: a common misconception is that the phase out in the US was voluntary. It was not. The drug manufacturers voluntarily removed the AGP label from the medically important drugs, which made it illegal to use these drugs for AGP in the US. For accuracy you could add something to the sentence like “currently there are no allowed uses of medically important AGPs, based on the drug labels.”

Line 99: delete second “administration”.

Line 155: last “AAI” on the line should have the “s” deleted.

Line 159-161: this calculation is useless and misleading. Why would anyone combine drug amounts for compounds with entirely different molecular weights and potencies? This makes no sense, even though many do it. I would suggest deleting this sentence and all totals in the paper that aggregate antimicrobials. I mentioned this in my last review. The authors do not seem to understand that drugs have different potency such that 1g of drug X is not the same as 1g of drug Y, so without an adjustment by potency, why would anyone combine these totals across drug classes?

Line 231: should say “commercial”

Lines 293-295: many mistakes with the English in this sentence. Fix.

Line 297: Should say “led”

Line 300: why do the authors think the data are accurate? The estimated drug amounts were taken from drug labels on the feed bags. Many of the bags had no labels for drug concentration. Given that many of the feeds were using illegal drugs or drug amounts, why do the authors assume that the drug labels themselves are accurate with respect to drug concentration? This should be restated and perhaps indicate that there is uncertainty due to labeling problems.

7. PLOS authors have the option to publish the peer review history of their article (what does this mean?). If published, this will include your full peer review and any attached files.

Reviewer #1: No

Reviewer #3: **Yes: **Jeroen Dewulf

Reviewer #4: No

---

## [Author Response · Author response to Decision Letter 1]

21 Mar 2021

Reviewer #1: lines 252-254: According to the authors, they have slightly reworded this sentence as follows: “There are very limited data on the contribution of AGPs in relation on total AMU in animal production. In Great Britain,---” although the first sentence is not identical to the revised manuscript. Should this part read “Much of the debate on AMU in animals has often been framed in terms of bans on AGPs. Unfortunately, global data on total amounts of AGPs consumed or on the contribution of AGPs on total AMU are lacking” in lines 252-254 of the revised MS? If so, this reviewer has no additional comment.

Response – The text has been now re-worded exactly as the reviewer suggests. 

lines 163-167: If the number of antimicrobial-containing feed products intended for chickens is thirty-five (line 163), description about the breakdown (lines 164-166) is incorrect. According to the S1 Table, twenty-one (60%, 21/35), 9 (25.7%, 9/35), and 5 (14.3%, 5/35) products are for brooding, growing, and finishing, respectively.

Response – Here we intended to reflect the proportion of feed products of each type that contained antimicrobials, not the distribution of products by purpose. We spotted, however a mistake in the calculations. In order to avoid confusion we have re-worded this sentence as follows: ‘A total of 21/38 (55.3%), 9/22 (40.9%) and 5/25 (20%) commercial feeds intendend for brooding, growing and finishing, respectively, contained antimicrobials’. 

S1 Table1: The title should include the term “chicken.” For example, Detailed information on antimicrobial-containing feed products intended for chickens.

Response – This have now been amended. 

Reviewer #4: The paper still has errors in grammar and syntax. This is unfortunate because PLoS does no copyediting, one reason not to publish in PLoS. Please proof the entire manuscript.

Response – We have re-read the manuscript with the help of native English speakers. We have amended a number of typos and grammar errors. We hope the manuscript is acceptable now. 

Remaining concerns:

Line 48: The authors did not address my concern regarding the statement that all EU countries “have well-developed antimicrobial consumption surveillance.” There is nothing well-developed about an AMU surveillance system that is completely based on sales data. It does not matter that ESVAC uses the term “consumption” in the title of the report. The fact is that the EU surveillance is based on sales data, and the countries do not have any information that would be important for characterizing AMU in animals. Most importantly, there are no data regarding the indication for use (disease). Of course, there also no data on dose, duration, number of animals exposed, etc. The ESVAC reports briefly mention these pitfalls. While AMU surveillance using sales data is an easy target for many countries, it should not be considered well-developed, complete, ideal or even useful for assessing whether uses are responsible and whether stewardship principles are being followed.

Response – We agree on your comment and have therefore amended this sentence: ‘In European Union (EU) countries, all of which have antimicrobial consumption surveillance systems based on sales data,…’

Line 50: Define AMU here.

Response – We have now spelled AMU here (‘antimicrobial use’).

Line 58: Remove AMU definition

Response – This has now been amended.

Line 70: a common misconception is that the phase out in the US was voluntary. It was not. The drug manufacturers voluntarily removed the AGP label from the medically important drugs, which made it illegal to use these drugs for AGP in the US. For accuracy you could add something to the sentence like “currently there are no allowed uses of medically important AGPs, based on the drug labels.”

Line 99: delete second “administration”.

Response – This has now been amended.

Line 155: last “AAI” on the line should have the “s” deleted.

Response – This has now been amended.

Line 159-161: this calculation is useless and misleading. Why would anyone combine drug amounts for compounds with entirely different molecular weights and potencies? This makes no sense, even though many do it. I would suggest deleting this sentence and all totals in the paper that aggregate antimicrobials. I mentioned this in my last review. The authors do not seem to understand that drugs have different potency such that 1g of drug X is not the same as 1g of drug Y, so without an adjustment by potency, why would anyone combine these totals across drug classes?

Response – We understand and agree with this comment. We are now presenting results by antimicrobial separately. 

Line 231: should say “commercial”.

Response – This has now been amended.

Lines 293-295: many mistakes with the English in this sentence. Fix.

Response – This has now been ameded, and reads like: ‘Indeed, recent studies in industrial (short cycle) broiler production systems showed that AGPs added to feed did not overall improve flock bodyweight [37, 38].’

Line 297: Should say “led”

Response – This has now been amended.

Line 300: why do the authors think the data are accurate? The estimated drug amounts were taken from drug labels on the feed bags. Many of the bags had no labels for drug concentration. Given that many of the feeds were using illegal drugs or drug amounts, why do the authors assume that the drug labels themselves are accurate with respect to drug concentration? This should be restated and perhaps indicate that there is uncertainty due to labeling problems.

Response – We acknowledge this limitation and thus have qualified this statement. Now it reads: ‘This study provides preliminary quantification on consumption of AGPs in native chicken production in the country based on examination of feed products’ labels. However, this information needs however to be taken with caution since a certain degree of inaccuracy in the AAI strength in feed labels is expected.

---

## [Decision Letter · Decision Letter 2]

31 Mar 2021

Antimicrobial use through consumption of medicated feeds in chicken flocks in the Mekong Delta of Vietnam: a three-year study before a ban on antimicrobial growth promoters

PONE-D-20-36867R2

Dear Dr. Carrique-Mas,

We’re pleased to inform you that your manuscript has been judged scientifically suitable for publication and will be formally accepted for publication once it meets all outstanding technical requirements.

Kind regards,

Iddya Karunasagar

Academic Editor

PLOS ONE

Additional Editor Comments (optional):

All reviewer comments have been addressed satisfactorily.

Reviewers' comments:

Reviewer's Responses to Questions

**Comments to the Author**

1. If the authors have adequately addressed your comments raised in a previous round of review and you feel that this manuscript is now acceptable for publication, you may indicate that here to bypass the “Comments to the Author” section, enter your conflict of interest statement in the “Confidential to Editor” section, and submit your "Accept" recommendation.

Reviewer #1: All comments have been addressed

Reviewer #4: All comments have been addressed

2. Is the manuscript technically sound, and do the data support the conclusions?

Reviewer #1: Yes

Reviewer #4: Yes

3. Has the statistical analysis been performed appropriately and rigorously? 

Reviewer #1: N/A

Reviewer #4: Yes

4. Have the authors made all data underlying the findings in their manuscript fully available?

Reviewer #1: Yes

Reviewer #4: Yes

5. Is the manuscript presented in an intelligible fashion and written in standard English?

Reviewer #1: Yes

Reviewer #4: Yes

6. Review Comments to the Author

Reviewer #1: (No Response)

Reviewer #4: Thank you for addressing my concerns. I cannot submit this review until 100 characters have been reached so I am just typing for the sake of typing.

7. PLOS authors have the option to publish the peer review history of their article (what does this mean?). If published, this will include your full peer review and any attached files.

Reviewer #1: No

Reviewer #4: No

---

## [Editor Report · Acceptance letter]

5 Apr 2021

PONE-D-20-36867R2 

Antimicrobial use through consumption of medicated feeds in chicken flocks in the Mekong Delta of Vietnam: a three-year study before a ban on antimicrobial growth promoters 

Dear Dr. Carrique-Mas:

I'm pleased to inform you that your manuscript has been deemed suitable for publication in PLOS ONE. Congratulations! Your manuscript is now with our production department. 

Kind regards, 

on behalf of

Dr. Iddya Karunasagar 

Academic Editor

PLOS ONE